# Kidney Involvement in Patients with Type 1 Autoimmune Pancreatitis

**DOI:** 10.3390/jcm8020258

**Published:** 2019-02-18

**Authors:** Miroslav Vujasinovic, Raffaella Maria Pozzi Mucelli, Roberto Valente, Caroline Sophie Verbeke, Stephan L. Haas, J.-Matthias Löhr

**Affiliations:** 1Department for Digestive Diseases, Karolinska University Hospital, SE-141 86 Stockholm, Sweden; roberto.valente@sll.se (R.V.); stephan.haas@sll.se (S.L.H.); matthias.lohr@ki.se (J.-M.L.); 2Department of Medicine, Huddinge, Karolinska Institute, SE-171 77 Stockholm, Sweden; 3Department of Abdominal Radiology, Karolinska University Hospital, SE-141 86 Stockholm, Sweden; raffaella.pozzi-mucelli@sll.se; 4Department of Clinical Science, Intervention, and Technology (CLINTEC), Karolinska Institute, SE-171 77 Stockholm, Sweden; 5Department of Pathology, Karolinska University Hospital, SE-141 86 Stockholm, Sweden; c.s.verbeke@medisin.uio.no; 6Department of Pathology, University Hospital of Oslo, Oslo 0450, Norway

**Keywords:** autoimmune pancreatitis, chronic, pancreatitis, kidney, tubulointerstitial nephritis, immunoglobulin G4

## Abstract

Introduction: Autoimmune pancreatitis (AIP) type 1 is a special form of chronic pancreatitis with a strong lymphocytic infiltration as the pathological hallmark and other organ involvement (OOI). IgG4-related kidney disease (IgG4-RKD) was first reported as an extrapancreatic manifestation of AIP in 2004. The aim of the present study was to determine the frequency and clinical impact of kidney lesions observed in patients with AIP type 1. Methods: We performed a single-centre retrospective study on a prospectively collected cohort of patients with a histologically proven or highly probable diagnosis of AIP according to the International Consensus Diagnostic Criteria (ICDC) classification. Results: Seventy-one patients with AIP were evaluated. AIP type 1 was diagnosed in 62 (87%) patients. Kidney involvement was present in 17 (27.4%) patients with AIP type 1: 15 (88.2%) males and 2 (11.8%) females. Laboratory and/or imaging signs of kidney involvement were presented at the time of AIP diagnosis in eight (47.1%) patients. In other patients, the onset of kidney involvement occurred between four months and eight years following diagnosis. At the time of the diagnosis of kidney involvement, eight (47.1%) patients showed elevated creatinine, and nine (52.9%) patients showed normal serum creatinine. None of the patients were treated with dialysis. Conclusions: IgG4-RKD was present in 27.4% of patients with AIP type 1, with male gender predominance. In cases of early diagnosis and cortisone treatment, the clinical course was mild in most cases. Regular laboratory control of renal function should be a part of the follow-up of patients with AIP type 1.

## 1. Introduction

Autoimmune pancreatitis (AIP) is a particular form of chronic pancreatitis with a heavy lymphocytic infiltration as the pathological hallmark and two distinct histopathological subtypes: Lymphoplasmacytic sclerosing pancreatitis (LPSP, AIP type 1) and idiopathic duct-centric pancreatitis (IDCP, AIP type 2) [1]. AIP is part of the IgG4-related diseases (IgG4-RDs) incorporating a wide range of other organ involvement (OOI). Among them, immune-associated cholangitis (IAC) is the most frequent [2,3].

Diagnosis of AIP is not always easy, and requires a combination of different clinical, laboratory and imaging data. According to the International Consensus Diagnostic Criteria (ICDC), the diagnosis of AIP is based on the presence of one or more of the following factors: pancreatic parenchyma and pancreatic duct imaging, serum IgG4 level, other organ involvement, histology of the pancreas and response to steroid treatment [4].

We recently reported our cohort of AIP patients [3]. OOI was present in 84% of AIP type 1 patients, which represents a higher prevalence rate compared to cohorts of other European studies varying from 47% to 61% [3,5,6,7,8]. The most common among OOI was cholangitis, followed by nephritis, inflammatory bowel disease, autoimmune hepatitis, retroperitoneal fibrosis, sialo adenitis, autoimmune thyroiditis, vasculitis, dacryoadenitis, duodenal papilla IgG4 involvement and lung involvement. Other unrelated autoimmune disorders were present as well, such as rheumatic polymyalgia, Sjogren’s syndrome, coeliac disease, psoriasis, autoimmune haemolytic anaemia and autoimmune gastritis. AIP is a treatable form of chronic pancreatitis with a good initial response to steroid therapy and relapse occurrence varies from 7% to 55% [2]. IgG4-related kidney disease (IgG4-RKD) was first reported as a complication or an extrapancreatic manifestation of AIP in 2004 [9,10]. In the early reports, patients showed renal impairment and/or proteinuria, and a renal biopsy revealed tubulointerstitial nephritis (TIN) and fibrosis with dense infiltration of IgG4-positive plasma cells. Thereafter, incidentally detected IgG4-RKD cases were described in the course of AIP or chronic sclerosing sialadenitis and dacryoadenitis, and even cases of isolated IgG4-RKD without AIP or chronic sclerosing sialadenitis and dacryoadenitis [11,12,13]. Most of the cases reported so far came from Asia (Japan). Thus, the Japanese Society of Nephrology (JSN) proposed a diagnostic algorithm and the diagnostic criteria for IgG4-RKD [14].

Here, we take a closer look at the prevalence of kidney lesions in patients with AIP type 1 in a Scandinavian cohort of patients.

## 2. Patients and Methods

### 2.1. Study Design

We performed a single-centre retrospective study on a prospectively collected cohort of patients seen at the outpatient clinic of the Department for Digestive Diseases at Karolinska University Hospital in Stockholm, Sweden from 2004 to 2018.

### 2.2. Cohort

Consecutive patients with a histologically proven or highly probable diagnosis of AIP, according to the ICDC, were included in the study. A retrospective analysis and diagnosis according to the ICDC were performed for patients diagnosed in the period before the publication of the ICDC [4].

Serum IgG4 levels between 0.05 and 1.25 g/L were considered as normal. Those IgG4 levels exceeding 1.25 g/L were deemed to be elevated.

Remission of AIP was defined as the disappearance of symptoms and imaging manifestations after the initial treatment. Relapse of AIP was defined as the recurrence of symptoms of AIP after initial resolution and/or radiological signs in the pancreas or extra-pancreatic organs after exclusion of other diseases [8].

IgG4-RKD was diagnosed according to criteria from the JSN as definite IgG4-RKD and suspected IgG4-RKD (Figure 1—Used with author’s approval) [14].

### 2.3. Imaging

The available CT and MRI examinations were evaluated on a picture archiving and communication system (PACS) (Sectra AB, Linköping, Sweden), by a dedicated abdominal radiologist. Since our department represents a tertiary care centre, we also included CT and/or MRI scans from other centres in Sweden.

In accordance with previously published proposals [14,15], the following features were recorded from imaging: (1) Administration of iodinated contrast agent; (2) unilateral or bilateral kidney involvement; (3) presence of a solitary hypodense lesion; (4) presence of multiple hypodense lesions; (5) diffuse bilateral renal swelling on unenhanced and/or enhanced CT; (6) diffuse thickening of the renal pelvis wall; (7) soft tissue mass in the perinephric space.

For MRI examinations, the following imaging aspects were documented: (1) Administration of gadolinium-based contrast agent; (2) unilateral or bilateral kidney involvement; (3) presence of a solitary lesion; (4) presence of multiple lesions; (5) diffuse bilateral renal swelling on unenhanced MRI sequences and/or enhanced MRI sequences; (6) diffuse thickening of the renal pelvis wall; (7) soft tissue mass in the perinephric space. Furthermore, the type of signal intensity (SI) (defined as hypointense/isointense/hyperintense compared to the surrounding normal renal parenchyma) on T2- and T1-weighted images before and after contrast medium administration, and the presence of restricted diffusion (defined as lesion hyperintensity at diffusion-weighted images (DWIs) with high b-values and hypo intensity on apparent diffusion coefficient (ADC) maps compared to the liver) were also recorded.

### 2.4. Ethics

The study was approved by the Clinic Ethical Committee in Stockholm (2016/1571-31) and adhered to the Declaration of Helsinki.

## 3. Results

Seventy-one patients with AIP were evaluated at Karolinska University Hospital between 2004 and 2018: 49% males with a mean age of 49 years (44–53). AIP type 1 was diagnosed in 62 (87%) patients. Kidney involvement was present in 17 (27.4%) patients with AIP type 1: 15 (88.2%) males and 2 (11.8%) females. The mean age of patients with kidney involvement at the time of diagnosis of AIP was 60.6 ± 13.1 years (range 39–85) and the mean age at the time of the data analysis was 64.6 ± 13.7 years (range 40–87). The mean time interval between the diagnosis of AIP and the data analysis was 3.9 ± 2.4 years (range 1–8). IgG4 values were elevated in 10 (58.8%) patients. The mean IgG4 value in all 17 patients was 4.0 g/L (range 0.3–20.7 g/L). The mean IgG4 levels in patients with elevated values were 6.4 g/L (range 1.4–20.7 g/L). According to the JNS criteria, definite IgG4-RKD was diagnosed in only 1 (5.9%) patient with available kidney histology, while the other 16 (94.1%) patients were diagnosed as suspected IgG4-RKD.

At the time of AIP diagnosis, creatinine values were elevated in two (11.8%) male patients: A 73-year-old (creatinine 120 µmol/L; glomerular filtration rate (GFR) 48) and a 68-year-old (118 µmol/L; GFR 50). The mean serum creatinine values in all 17 patients were 82.5 µmol/L (range 60–120 µmol/L), and the mean GFR was 81.4 ± 21.0 (range 48–129).

During the observational period, eight (47.1%) patients had elevated creatinine levels. At the time of data analysis (final contact with the patients) elevated creatinine values were present in five (29.4%) patients (all males): An 81-year-old (creatinine 122 µmol/L; GFR 43), a 48-year-old (183; 33), a 76-year-old (130; 43), a 73-year-old (117; 50) and an 87-year-old (101; 50). A flowchart of patients is presented in Figure 2.

Laboratory and/or imaging signs of kidney involvement were present at the time of diagnosis in eight (47.1%) patients. In the other nine (52.9%) patients the onset of kidney involvement occurred after 4 months, 5 months, 6 months, 7 months, 11 months, 2 years, 4 years, 5 years and 8 years, respectively. Haematuria was present in four (23.5%) patients, and proteinuria was also present in four patients (in one patient, both haematuria and proteinuria were present). OOI (besides pancreas and kidney) was present in 16 (94.1%) patients (cholangitis, vasculitis, abdominal lymph node swelling, lung involvement, retroperitoneal fibrosis, autoimmune hepatitis, Sjögren’s Syndrome and mediastinal lymph node swelling).

A total of 13 patients (76.5%) underwent treatment because of AIP, of whom 12 (70.6%) received cortisone, 1 (5.9%) azathioprine, 1 (5.9%) rituximab, 2 (11.8%) biliary stenting because of a stricture and 3 (17.6%) underwent surgery (patients in whom it was not possible to differentiate AIP from pancreatic carcinoma). Five (29.4%) patients received a combination of treatments and eight (47.1%) underwent monotherapy. After initial treatment with cortisone, all patients responded well, but 12 (70.6%) patients relapsed AIP. According to the status of AIP, 11 (64.7%) patients were in complete clinical remission without therapy at the time of data analysis, 1 (5.9%) was in remission on treatment and 5 (29.4%) took medication due to relapse. According to serum creatinine values at the time of the last contact, 10 (58.8%) patients showed normal creatinine, 3 (17.6%) had normal creatinine with imaging signs of kidney lesions and 4 (23.5%) had chronic kidney disease stage 3. None of the patients were treated with dialysis. There were no active smokers, but four (23.5%) patients were former smokers. None of the patients had high alcohol consumption (>140 g per week). Diabetes mellitus (DM) was present in seven (41.2%) patients: In three patients before AIP diagnosis and in four patients after AIP diagnosis and treatment with surgery and steroids. Arterial hypertension (AH) was present in five (29.4%) patients: In three patients before AIP diagnosis and in two patients after AIP diagnosis.

Demographic, clinical and radiological characteristics of individual patients are presented in Table 1.

### Imaging Features

An abdominal CT scan with intravenous contrast agent was available for 11 (64.7%) patients. In nine cases, the examination consisted of an arterial and venous phase. In one case, a delayed phase was also acquired. In one patient, only venous phase acquisition was available.

An MRI of the upper abdomen was available for all 17 patients (in 16 cases, it was performed with the injection of gadolinium-based contrast agent). However, in one case (patient number 14 in Table 1) the very low image quality did not allow for proper evaluation. Thus, this MRI was excluded, and 16 MRIs were analysed.

The analysis of the CT scans revealed a bilateral kidney involvement in 10 (90.9%) patients, with multiple hypodense lesions in 8 out of 10 patients (90%), presence of a soft tissue mass in the perinephric space with bilateral diffuse renal swelling on both unenhanced and enhanced in one case (1/10, 10%), and bilateral focal thinning or absence of the renal cortex in the remaining case (1/10, 10%). In one case (1/11, 9.1%), the CT revealed a unilateral kidney involvement with a solitary hypodense lesion affecting the right kidney. No patients showed diffuse thickening of the renal pelvis wall.

The analysis of MRIs revealed a bilateral kidney involvement in 13 cases (13/16, 81.2%), with multiple hypodense lesions in 11 cases (11/13, 84.6%), focal thinning or absence of renal cortex in 1 case (1/13, 7.7%) and presence of a bilateral soft tissue mass in the perinephric space in the remaining patient (1/13, 7.7%). In one case, a unilateral kidney involvement with multiple focal lesions was observed. In two patients, unilateral kidney involvement with a solitary lesion affecting the right kidney was recorded. Demographic, clinical and radiological characteristics of individual patients are presented in Table 1. The SI of single and/or multiple focal lesions in the different sequences is reported in Table 2.

## 4. Discussion

In most patients with AIP type 1, other organ involvement (OOI) occurred during the clinical course, sometimes simultaneously and often metachronously [16,17]. According to our recently published single-centre data, OOI was present in 84% of patients with AIP with cholangitis as the most common associated condition, followed by inflammatory bowel disease, nephritis, autoimmune hepatitis, retroperitoneal fibrosis, sialadenitis, autoimmune thyroiditis, vasculitis, dacryoadenitis, duodenal papilla IgG4-involvement, lung involvement and lymphadenopathy of mediastinal and abdominal lymph nodes [3].

Renal parenchymal lesions associated with IgG4-related disease were first described in 2004 as case reports of tubulointerstitial nephritis associated with AIP [9,10]. Saeki et al. described a cohort of 23 patients with renal parenchymal lesions associated with IgG4-related disease with characteristic clinicopathological features and the term “IgG4-related tubulointerstitial nephritis” was proposed for this condition [18]. Kawano et al. described a well-characterised cohort of 41 patients with IgG4-RKD, and the authors proposed a diagnostic algorithm and diagnostic criteria for IgG4-RKD: Clinical features including extra-renal organ involvement, urinalysis and serological features including serum IgG4 levels, imaging findings based on CT, renal histology with IgG4 immunostaining and response to steroid therapy [14]. According to the criteria mentioned above, most of our patients (12 out of 17) were classified as suspected IgG4-RKD, as pancreatic histology was only available for four patients and a renal biopsy was performed in one patient (Figure 1). A moderate to marked increase in IgG4 plasma cells is one of the diagnostic criteria for AIP in the pancreas. Nevertheless, such an increase was diagnostically helpful when present, but only 70%–72% of AIP patients had at least moderately increased IgG4 plasma cells in pancreatic specimens [19,20].

To avoid unnecessary invasive diagnostic procedures, renal biopsy was not performed, and in most cases conservative treatment with cortisone was introduced, including regular laboratory follow-up of creatinine values. After initial treatment with cortisone, all patients responded well, but relapse of AIP occurred in 12 (70.6%) patients. However, kidney function was not significantly impaired. During the observational period, eight (47.1%) patients had elevated creatinine values and at the time of data analysis, creatinine values were elevated in only two (11.8%) male patients: A 73-year-old (creatinine 120 µmol/L; GFR 48) and a 68-year-old (118 µmol/L; GFR 50), suggesting a mild clinical course. Mean serum creatinine values in all 17 patients was 82.5 µmol/L (range 60–120 µmol/L) and mean GFR was 81.4 ± 21.0 (range 48–129).

Laboratory and/or imaging signs of kidney involvement were present at the time of diagnosis in eight (47.1%) patients. In the other nine patients, the onset of kidney involvement occurred later (between 4 months and 8 years), suggesting that regular laboratory control of kidney function should be a part of the follow-up of patients with AIP type 1, which is already routinely performed at our centre.

The average age of our patients was 60.6 years with a male gender predominance, similar to studies from Japan (Table 3). Elevated serum creatinine values in half of the patients and mild clinical course of renal failure were also similar. There was only one patient from each Japanese study who was treated with haemodialysis, whereas no patients in our study received this treatment.

In chronic kidney disease, albuminuria is a recognized indicator of renal function and assessed along with GFR, and urine albumin/creatinine ratio (uACR) is recommended as a simple measure of albuminuria [21]. However, uACR was known in only 5 out of 17 patients in our study (4 with normal values and 1 elevated) and we were not able to provide more data.

In a study by Saeki et al., 14 of the 23 patients with kidney involvement were without pancreatic lesions, however, the clinical features were rather uniform and similar to those shown in AIP. Their results suggested that renal parenchymal lesions developed in association with IgG4-related disease, but not in association with AIP [18]. Recently, serum angiopoietin-2 (Ang-2) has been associated with hyperdynamic state of the systemic circulation in patients with acute pancreatitis (AP) and proposed as a relevant predictor of AP, in particular of the development of AP-renal syndrome [21]. Another promising biomarker for acute kidney injury is neutrophil gelatinase-associated lipocalin (NGAL) [22]. However, only 22% of patients in our initial cohort of AIP patients presented with acute pancreatitis and the rest of the patients were diagnosed with radiological signs of chronic pancreatitis, with 47% of them with pancreatic exocrine insufficiency [3].

Tubulointerstitial nephritis can be triggered by a range of different aetiologies and can be immunologically mediated, associated with drugs (allergic or toxic), associated with infection (either by direct infection or in reaction to a distant infection), hereditary, metabolic or may be due to other causes or overlap between categories [23,24]. DM and AH were probably not important aetiological factors in our patients: DM was presented in seven (41.2%) patients (in three patients before AIP diagnosis and four patients after AIP diagnosis and treatment with surgery and steroids) and AH was presented in five (29.4%) patients (in three patients before AIP diagnosis and two patients after the AIP diagnosis). There were no active smokers, but four (23.5%) patients were former smokers.

Another important consideration is about the valuable contribution of MRI to the evaluation of patients with IgG4-RKD. As previously described by Kim B. et al. [15], DWI is an essential tool for the detection of renal lesions. In our series, DWI could detect renal parenchymal lesions as hyperintense areas in 93.4% of the cases, while these lesions were isointense, and thus not detectable, in 40%–46.6% of the cases in the dynamic sequences after contrast agent. Hence, MRI with DWI may be safely used without the injection of gadolinium-based contrast agent for the detection and/or follow-up of IgG4-RKD, especially in case of an impaired renal function (i.e., avoiding the risk of developing nephrogenic systemic fibrosis).

The main limitation of the present study is its retrospective nature. Nevertheless, it describes a relatively high number of patients, considering the rarity of the disease, helping to shed light on an almost unexplored area of the management of IgG4-RKD in patients with AIP type 1. Additionally, to the best of our knowledge, the current study represents the first relatively large Western study on the issue and is the first to describe clinical/radiological characteristics and long-term outcomes.

Seventeen out of 62 (27.4%) patients in our cohort of type I AIP developed kidney involvement from an IgG4-RKD and 10/17 (58.82%) developed kidney failure. A close monitoring of kidney functionality is recommended in patients with type I AIP and radiological signs of kidney involvement, and treatment should be considered to avoid the development of definitive kidney failure. Further, possibly multicentre double-blind control trials are needed to assess the long-term effect of immunosuppressive drugs on long-term kidney functionality.

## 5. Conclusions

IgG4-RKD was present in every fourth (27.4%) patient with AIP type 1. Patients had a male gender predominance and were approximately 60 years of age. When diagnosed in the early course and treated with cortisone, the clinical course was mild in most of the cases. Regular laboratory control of kidney function should be a part of the follow-up of patients with AIP type 1.

## Figures and Tables

**Figure 1 jcm-08-00258-f001:**
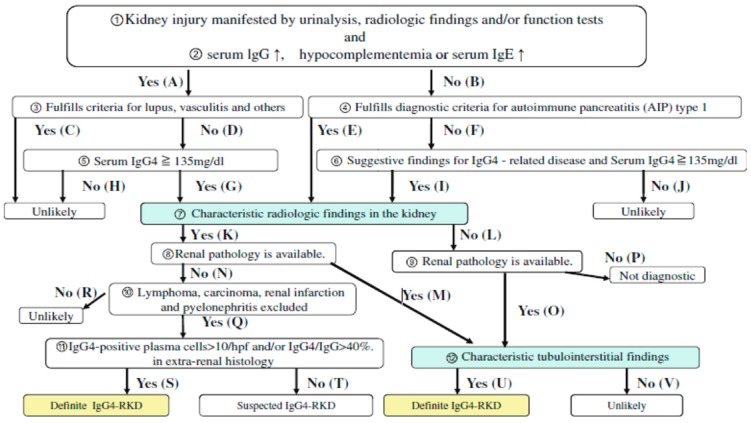
Diagnostic algorithm for IgG4-related kidney disease (IgG4-RKD) as proposed by Kawano et al. (with permission) [14].

**Figure 2 jcm-08-00258-f002:**
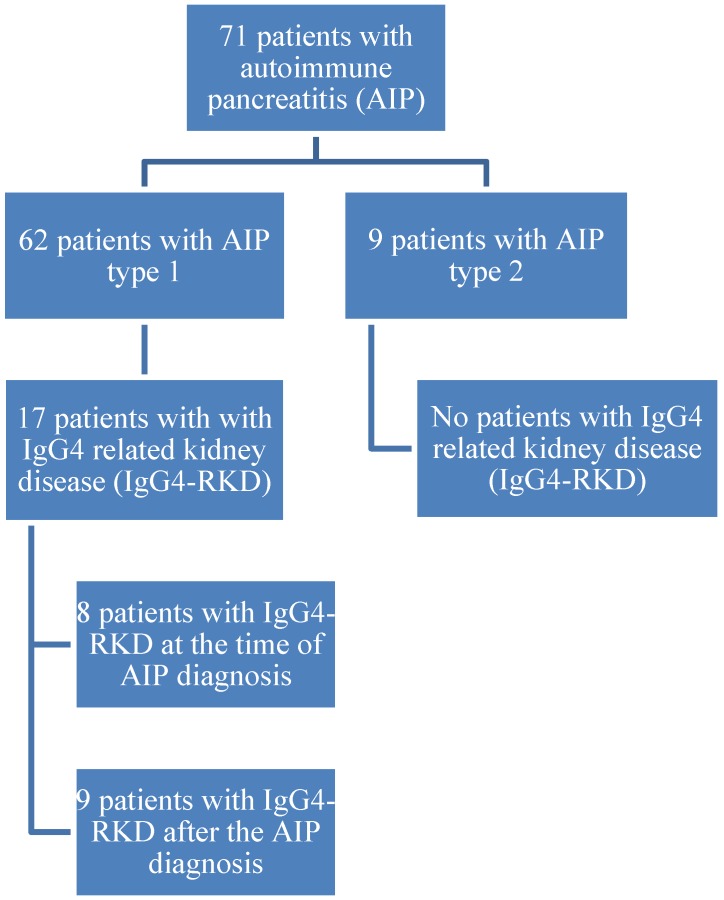
Flowchart of patients.

**Table 1 jcm-08-00258-t001:** Demographic, clinical and radiological characteristics of individual patients.

N	Gender	Age	Treatment	OOI (Other than Kidney/Pancreas)	Imaging	Type of Kidney Involvement	Unilateral vs. Bilateral Involvement	Onset of Kidney Involvement
1	M	74	steroids, surgery	cholangitis	CEMR	multiple lesions	bilateral	6 y after AIP
				vasculitis (aorta)				
				retroperitoneal fibrosis				
2	F	73	steroids	cholangitis	CECT CEMR	multiple lesions	bilateral	3 m after AIP
				Sjögren’s Syndrome enlarge mediastinal LN				
3	M	52	steroids, biliary stent	cholangitis	CECT MRw/o c	multiple lesions	bilateral	synchronous
4	M	49	steroids, surgery	cholangitis	CECT CEMR	soft tissue in the perinephric space, diffuse swelling	bilateral	6 m after AIP
				hepatitis				
				enlarge abdominal LN				
5	M	60	steroids	cholangitis	CEMR	multiple lesions	unilateral (left)	11 m after AIP
6	M	57	steroids, azathioprine	cholangitis	CECT CEMR	solitary lesion	unilateral (right)	synchronous
				enlarge abdominal LN vasculitis (aorta)				
7	M	42	steroids	cholangitis	CECT CEMR	multiple lesions	bilateral	synchronous
8	F	39	none	cholangitis	CECT CEMR	multiple lesions	bilateral	synchronous
				lung involvement				
9	M	39	none	cholangitis	CECT CEMR	multiple lesions	bilateral	synchronous
10	M	73	none	cholangitis	CEMR	multiple lesions	bilateral	synchronous
11	M	68	steroids	none	CECT CEMR	multiple lesions	bilateral	synchronous
12	M	68	steroids, surgery	cholangitis	CECT CEMR	focal thinning of renal cortex	bilateral	synchronous
13	M	85	biliary stent	cholangitis	CECT CEMR	multiple lesions	bilateral	synchronous
				lung involvement				
				vasculitis (aorta)				
14	M	71	steroids	cholangitis	CECT *	multiple lesions	bilateral	2 y after AIP
15	M	65	steroids	cholangitis	CEMR	multiple lesions	bilateral	8 y after AIP
				vasculitis (aorta)				
16	M	52	none	cholangitis	CEMR	multiple lesions	bilateral	4 y after AIP
17	M	64	none	cholangitis	CEMR	solitary lesion	unilateral (right)	synchronous

Abbreviations: M = male; F = female; AIP = autoimmune pancreatitis; OOI = other organ involvement; LN = lymph nodes; CECT = contrast-enhanced computed tomography; CEMR = contrast-enhanced magnetic resonance; MRw/o c = MR without contrast agent; y = years; m = months. * MR in patient 14 was excluded from the data analysis due to the low image quality. Age at the time of diagnosis.

**Table 2 jcm-08-00258-t002:** Signal intensity (SI) at MRI for single and/or multiple focal kidney lesions, compared to the surrounding normal parenchyma.

Signal Intensity (SI)	MRI Sequences					
	T2-Weigthed	T1-Weighted (w/o Contrast Agent)	T1-Weighted Arterial Phase	T1-Weighted Venous Phase	T1-Weighted Delayed Phase	DWI *
Hypointense	10/15 (66.6%)	2/15 (13.4%)	9/15 (60%)	8/15 (53.4%)	9/15 (60%)	1/15 (6.6%)
Isointense	5/15 (33.4%)	13/15 (86.6%)	6/15 (40%)	7/15 (46.6%)	6/15 (40%)	0
Hyperintense	0	0	0	0	0	14/15 (93.4%)
Restricted SI	-	-	-	-	-	10/11 (90.9%)

Sixteen MRIs were evaluated (one MRI was excluded due to very low image quality). In one patient (subject 12), SI analysis was not performed because of the presence of multiple areas of focal thinning of the renal cortex. Thus, data from only 15 patients are summarised in this table. * Diffusion-weighted images (DWIs) were available in 15/16 MRIs (not acquired in patient 12).

**Table 3 jcm-08-00258-t003:** Comparison of our results with studies from Japan.

Parameter	Present Study	Saeki et al. [18]	Kawano et al. [14]
Number of patients	17	23	41
Gender	15 (88.2%) male and 2 (11.8%) female	20 (86.9%) male and 3 (13.1%) female	30 (73.2%) male and 11 (26.8%) female
Age at diagnosis (years)	60.6 ± 13.1 (range 39–85)	65.2 ± 10.1 (range 40–83)	63.7 ± 12.3 years (range 27–83)
OOI %	94.1	95.7	95.1
Haematuria %	23.5	34.8	41.7
Proteinuria %	23.5	8.7	58.3
Elevated creatinine values %	47.1	56.5	58.5
Elevated IgG4 values %	58.8	100	100
Treatment with corticosteroids %	76.5	91.3	92.7
Improvement after steroid therapy %	100	94.7	92.1
Haemodialysis after steroid therapy %	0	5.2	2.6

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
