# Peer review of "Kidney Involvement in Patients with Type 1 Autoimmune Pancreatitis"

_jcm, 2019, doi:10.3390/jcm8020258_

Round 1
Reviewer 1 Report
Review of the article: Kidney involvement in patients with type 1 autoimmune pancreatitis
The topic is very important and popular within pancreatology. Diagnosis, treatment and research of AIP is a challenge because of the low incidence of disease. Authors investigated the kidney involvement in patients who suffered from type I autoimmune pancreatitis.
The study is a retrospective, single-center study prospectively collected cohort of patients with histologically proven or highly probable diagnosis of AIP according to the ICDC classification. Seventy-one patients with AIP were evaluated. AIP type 1 was diagnosed in 62 (87%) patients. Kidney involvement was present in 17 (27.4%) patients with AIP type 1. Laboratory and/or imaging signs of kidney involvement were present at the time of AIP diagnosis in 8 (47.1%) patients. In other patients, the onset of kidney involvement occurred between 4 months and 8 years following diagnosis. At the time of the diagnosis of kidney involvement, 8 (47.1%) patients showed elevated creatinine, and 9 (52.9%) patients showed normal serum creatinine. IgG4-RKD is presented in 27.4% of patients with AIP type 1 with male gender predominance. In cases of early diagnosis and cortisone treatment, the clinical course is mild in most of the cases. Regular laboratory control of renal function should be a part of the follow-up of patients with AIP type 1.
The study is a well-designed, carried out research and well written paper. The part of introduction should be completed with more pieces of information about: AIP, IgG4-related diseases, IgG4-related kidney disease. Could you suggest a surveillance strategy for the diagnosis and treatment of IgG4-related kidney disease in the discussion or conclusion part of the study? Is it possible to carry out a statistical analysis to compare the data of recent study with former studies in the discussion part?
After answering the questions, the suggestion is minor revision.
Yours sincerely
Author Response
Reviewer 1:
“The study is a well-designed, carried out research and well written paper. The part of introduction should be completed with more pieces of information about: AIP, IgG4-related diseases, IgG4-related kidney disease. Could you suggest a surveillance strategy for the diagnosis and treatment of IgG4-related kidney disease in the discussion or conclusion part of the study? Is it possible to carry out a statistical analysis to compare the data of recent study with former studies in the discussion part?”
Authors’ answer:
Thank you very much for your valuable comments!
1 We completed and expanded introduction part of the manuscript according your instructions.
2 We suggested in conclusion part of the manuscript that “Regular laboratory control of kidney function should be a part of a follow up of patients with AIP type 1.” Even more, in discussion we emphasised that “In patients with type I AIP and radiological signs of kidney involvement a close monitoring of kidney functionality is recommended, and treatment should be considered to avoid the development of definitive kidney failure.”
3 Unfortunately, advanced statistics is not possible at the moment, due to the low number of the patients included in the studies and different selection of the patients that can be important bias (Saeki et al, and Kawano et al, included patients with histopathological confirmed IgG4 kidney disease as a basic group of patients and we included patients with autoimmune pancreatitis type 1 with laboratory and imaging signs of the kidney disease). However, your observation is of importance and we emphasised that “Further, possibly multicentre double-blind control trials are needed to assess the long-term effect of immunosuppressive drugs on long term kidney functionality.” As a surrogate to your proposal we included table 3 with direct comparison of the most important results in our study and two studies from japan so the readers can get information on this topic.
Reviewer 2 Report
The aim of the present study by Vujasinovic and other was to determine the frequency and the clinical impact of kidney lesions observed in patients with AIP type 1. It is interesting article. After reading this article submitted to me for review, however, it occurred to me observations and comments. The described comments and suggested changes in the text lead to a better understanding of the theme and will increase readers' interest in this topic. Here they are.
Point 1.
The section "Introduction" is too short and insufficient. In this section, complete lack of information on one of the most important etiological factors for the development of acute pancreatitis (AP) - pancreatic blood flow and the recent studies on the effects of coagulation activation (for example PMID: 28208708, PMID: 28368336). To better understand the topic should be extended introduction about the latest pancreatic hormone physiology (for example PMID: 25716961). It is also important to discuss the search for new prognostic factors (mainly predictive) in the course of AP - this topic is, after all, the most recent in recent pancreatology (for example PMID: 28067818, PMID: 28013317, PMID: 27929426, PMID: 29068376). This should be discussed. Such changes will result in the introduction of a better understanding of this topic.
Point 2.
In the "Discussion" section, reference should be made to studies on new indicators monitoring kidney function (for example Angiopoietin-2 (PMID: 27022209), urine NGAL PMID:27513835, PMID:28050059).
Point 3.
The authors do not present the albumin/creatinine ratio in presented and discussed research. This problem needs to be discussed (PMID:29324682). The authors should refer to research on this.
Author Response
Reviewer 2:
“Point 1. The section "Introduction" is too short and insufficient. In this section, complete lack of information on one of the most important etiological factors for the development of acute pancreatitis (AP) - pancreatic blood flow and the recent studies on the effects of coagulation activation (for example PMID: 28208708, PMID: 28368336). To better understand the topic should be extended introduction about the latest pancreatic hormone physiology (for example PMID: 25716961). It is also important to discuss the search for new
prognostic factors (mainly predictive) in the course of AP - this topic is, after all, the most recent in recent pancreatology (for example PMID: 28067818, PMID:28013317, PMID: 27929426, PMID: 29068376). This should be discussed. Such changes will result in the introduction of a better understanding of this topic.
Point 2.In the "Discussion" section, reference should be made to studies on new indicators monitoring kidney function (for example Angiopoietin-2 (PMID:27022209), urine NGAL PMID:27513835, PMID:28050059).
Point 3.The authors do not present the albumin/creatinine ratio in presented and discussed research. This problem needs to be discussed (PMID:29324682).”
Authors’ answer:
Thank you very much for your valuable comments!
Please find enclosed our answers:
Point 1:
We agree with your objections and we expanded the introduction section of the manuscript with additional information on autoimmune pancreatitis, other organ involvement, treatment of the diseases and the natural course.
Only 22% of patients in our cohort presented with acute pancreatitis and the rest of the patients were diagnosed with radiological signs of chronic pancreatitis with 47% of them with pancreatic exocrine insufficiency. That was the reason why we did not discuss deeply pathophysiological background of acute pancreatitis. We are very familiar with very nice preclinical studies on the most important etiological factors for the development of acute pancreatitis (AP) - pancreatic blood flow and with the recent studies on the effects of coagulation activation, however, the role of early pathologic events in AP (vascular derangements, endothelial activation and injury, dysregulation of vasomotor tone, increased vascular permeability, increased leukocyte migration to tissues, activation of coagulation, the role of angiopoietin-2 and soluble fms-like tyrosine kinase 1) were out of the scope in presenting article. The same is with the role of gastrointestinal hormones. Their connection with endocrine and exocrine function and the protection of the pancreas is well known but to our best knowledge they are not related in pathophysiology of AIP. We are thankful the reviewer for comments and we will try to take in consideration in the future work on this topic.
Point 2:
New indicators monitoring kidney function (angiopoietin-2 and urine NGAL) are important in the context of acute pancreatitis and acute kidney injury. We included this in the discussion part (new references are added as well).
Point 3:
We added comment on the importance of the albumin/creatinine ratio and we add the new reference.
Round 2
Reviewer 2 Report
The authors have completed recommendations. In my opinion, this article after the changes is good.